# Organisational Measures and Strategies for a Healthy and Sustainable Extended Working Life and Employability—A Deductive Content Analysis with Data Including Employees, First Line Managers, Trade Union Representatives and HR-Practitioners

**DOI:** 10.3390/ijerph18115626

**Published:** 2021-05-25

**Authors:** Kerstin Nilsson, Emma Nilsson

**Affiliations:** 1Division of Occupational and Environmental Medicine, Lund University, 223 81 Lund, Sweden; emma.nilsson.1672@med.lu.se; 2Department of Public Health, Kristianstad University, 291 88 Kristianstad, Sweden

**Keywords:** employability, ageing, senior, work environment, swAge-model, demography, retirement, global sustainable goal, public health, health promotion, health prevention, empowerment, occupational health care, psychosocial, working hours, recuperation, recovery, private finance, economy, work–life balance, manager, social support, discrimination, motivation, job satisfaction, competence, work ability, creativity, gerontology, older worker, extended working life, age management

## Abstract

Due to the global demographic change many more people will need to work until an older age, and organisations and enterprises need to implement measures to facilitate an extended working life. The aim of this study was to investigate organisational measures and suggestions to promote and make improvements for a healthy and sustainable working life for all ages in an extended working life. This is a qualitative study, and the data were collected through both focus group interviews and individual interviews that included 145 participants. The study identified several suggestions for measures and actions to increase employability in the themes: to promote a good physical and mental work environment; to promote personal financial and social security; to promote relations, social inclusion and social support in the work situation; and to promote creativity, knowledge development and intrinsic work motivation, i.e., based on the spheres of determination in the theoretical swAge-model (sustainable working life for all ages). Based on the study results a tool for dialogue and discussion on employee work situation and career development was developed, and presented in this article. Regular conversations, communication and close dialogue are needed and are a prerequisite for good working conditions and a sustainable working environment, as well as to be able to manage employees and develop the organisation further. The identified measures need to be revisited regularly throughout the employees’ entire working life to enable a healthy and sustainable working life for all ages.

## 1. Introduction

The average life expectancy is above 80 years of age in more than one in three countries on Earth [1,2,3]. In particular, the ageing of the population is expected to be very rapid in Greece, Korea, Poland, Portugal, Slovakia, Slovenia and Spain, while Japan and Italy are already among the countries with the oldest populations. The proportion of the population active in the labour force is ageing in many countries as well. The population of the labour force in the OECD countries (the Organization for Economic Co-operation and Development) is estimated to decrease by an average of ten percent in 2060 [2]. However, this varies between countries; in Greece, Japan, Korea, Latvia, Lithuania and Poland the proportion of working age population is estimated to decrease by 35 percent or more. With a greater number of pensioners that can no longer contribute to the proportion of hours worked in the countries’ economies the average old-age to working-age demographic ratio, computed by keeping age thresholds constant, will result in an increased economic burden for the individuals included in the labour force. In 2080, the number of people older than 65 years of age is estimated to be 58 per 100 people of working age in the OECD countries. To exemplify, this means that 1.7 people of the estimated working age (20–64 years of age) must support each pensioner in a pension system where the retirement age is 65 years of age. This can be compared, for example, with the fact that in 1950 in Sweden the old age dependency burden was about six people of working age (20–64 years of age) who would support every pensioner aged 65 and older. At the same time, the individuals in the labour force must also contribute to other facets in society, such as the support of children and young people who have not yet entered the work force and the individuals aged 20–64 who are not part of working life for other reasons. Rapid ageing of the population contributes to increased pressure on the pension systems in various countries to deliver an adequate and financially sustainable pension system and pensions. The financial crises that have occurred in several countries have also contributed to high public debt and limited scope for manoeuvring the national economy. This increases the risk of widening gaps in society and between countries caused by changes in the working environment, low growth and low interest rates, leading to new challenges for already strained pension systems in several countries. However, low interest rates generate both challenges and opportunities since low interest rates also mean low interest rates for countries’ public debt. In contrast, the challenges in ageing societies can also lead to increased benefit payments, which in turn can contribute to higher taxes, lower wage growth, higher unemployment as well as reduced pensions for pensioners. A greater number of people need to keep working to maintain the welfare system in a sustainable financial manner [1,2,3]. However, in society some hold the attitude that senior workers should quit working and give younger people and the unemployed a chance to enter the labour market. But since the labour market and economy are not static over time, a younger individual cannot automatically take over an older individual’s work when the latter retires. With more people staying in the workforce until an older age, purchasing power and demand for goods and services are also expected to increase. Furthermore, a greater number of elderly people will probably increase the need for health promotion efforts and health care since health risks increase with the degeneration of organs and minds, and elderly people make up a large proportion of those in need of care, resulting in increasing the costs of health care in different countries.

To achieve sufficient financial sustainability and to maintain pensions, many countries are postponing the retirement age so that a greater number of people can work for longer and thereby contribute to the national economy. This is because the labour force largely finances the non-working and elderly in the population. However, postponing the retirement age has often proven to be one of the more controversial reforms in many countries, if measures are not promoted at the same time in working life in order to enable a sustainable working life for all ages and to maintain employability. Individuals’ employability depends on, for example, their health, competences and ability to function at work but also perceived labour market values. The challenge is how to enable and incentivise people to remain in working life until an older age [1,2,3]. It is important that individuals maintain good employability until an older age to make it possible and effective to extend working life and postpone the pension age as a measure to promote a sound social economy and to support welfare systems in different countries [4,5,6,7,8,9,10,11]. 

Earlier studies and a theoretical model on sustainable working life for all ages and employability state nine determinant areas that influence whether people can and want to work and be part of the labour force until an older age, and to promote a healthy and sustainable working life for all ages [5,6,7,12,13,14,15]. The swAge model (sustainable working life for all ages) aims to visualise the complexity, organise and make the connections more understandable. The swAge model is a theoretical model consisting of nine different determinant areas that are significant to a sustainable working life for all ages and that relate to the four spheres of determination regarding employability, the possibility of being able to and willing to be part of working life and different ways of defining age:A.The health effects of the work environment, which relate to biological age and ageing and include the following areas of determination: (1) Self-rated health, diagnoses and functional diversity, (2) physical work environment with unilateral movements, heavy lifting, risk of accidents, climate, chemical exposure and risk of contagion, (3) mental work environment with risk of stress and fatigue syndrome, threats and violence, and (4) working hours, work pace and the possibility of recuperation during and between work shifts for the employee. A sufficiently sound health is a prerequisite for employability and to be included in working life. However, professional work also affects the biological ageing, the physical and mental health and the need for recovery based on the physical and mental stresses, the wear and tear resulting from work, but also by the strengthening impacts of our work.B.Economics and financial incentives, which relate to chronological ages’ association to society’s control of various financial carrots and sticks, for example, through the pension system and social insurance system. Economics and financial incentives include the following determinant area: (5) The personal financial situation’s effects on individuals’ needs and willingness to work. Professional work contributes to the upkeep of livelihood, food and living expenses, and is often the main source of funding for individuals’ lives. Issues with employability due to ill health, lack of support and lack of skills risk causing exclusion from working life and a poorer financial situation for the individual, not least in bad times, e.g., through sick leave, unemployment and early retirement.C.Relationships, social support and participation, i.e., attitudes in the social context in which the individual finds himself/herself, are considered as well as whether the individual feels included or excluded in the group and receives sufficient social support from the environment when needed, relate to social age and ageing during the life cycle and include the areas of determination: (6) The effects of the personal social environment, with family, friends and leisure context, and (7) the social work environment with leadership, discrimination and the significance of the employment relationship context for individuals’ work. Humans are herd animals and working life can contribute to the experience of participation and inclusion in a group, as well as a sense of security. In spite of this, working life can also contribute to the experience of exclusion and neglect, or even discrimination. However, every employee also has a personal life and factors in the environment outside of work, their personal relationships also affect the individual’s opportunities and willingness to work. This also affects the employability of the individual.D.Execution of tasks and activities relate to cognitive age and ageing, intelligence ability, memory, learning and instrumental support, and include the following areas of determination: (8) Motivation, satisfaction and stimulation in the execution of work tasks, and (9) knowledge, competence and the importance of competence development for the individual’s work. Working life is constantly changing and employees must be and remain employable in relation to the requirements of knowledge and skills in order to execute the activities and tasks that their work entails. The tasks and activities at work can be a source of motivation, stimulation and joy; we are challenged, learn new things and develop; however, it can also be a source of boredom, dissatisfaction and stagnation from which individuals want to make their exit as soon as possible.

However, the attitudes towards the extension of working life vary according to different levels in the societies. The organisations and enterprises need to implement sustainable measures and strategies to make an extended working life possible for a greater number of individuals [4,5,6,7,8,9,10,11,12,13,14,15,16]. It is therefore of great importance to discuss and explore organisational measures with the intent of supporting employability and enabling people to participate in working life for longer, since several countries and societies plan to postpone the retirement age. Hence, the demographic shift and its challenges and opportunities are of special interest to societies, and require the implementation of policies and measures by organisations and enterprises in order to help people stay healthy, active and employable until an older age. Based on prior knowledge and reviewing literature on the subject, there is a limited number of studies and information on how organisations and enterprises can and wish to implement measures and strategies to promote a healthy and sustainable extended working life.

The aim of this study was to investigate organisational measures and suggestions to make improvements for a healthy and sustainable working life for all ages, in an extended working life. 

## 2. Method

The research question, i.e., to investigate organisational measures and suggestions to make improvements for a healthy and sustainable working life for all ages in an extended working life, is one of complexity and involves different contexts in the organisations and enterprises. Therefore, the research design was decided to be qualitative and the data were collected by means of interviews, in order to provide an in-depth investigation of the organisational measures and strategies that can promote a healthy and sustainable working life until an older age. 

### 2.1. Study Population

To maximise the number of participants according to heterogeneity, the recruitment sites included individuals of different sex, different positions and professions, from work domains in both female- and male-dominated workplaces and both from the public as well as the private sector. The total study population consisted of 145 participants, including: first line managers, senior employees (55–72 years of age) in both blue- and white-collar professions, trade union workers, and human resources (HR) personnel (Table 1). The senior employee occupations were nurse, nurse assistant, physician, social worker, medical secretary, carpenter, construction worker, concrete worker, engineer, technician, mechanic, installer, electrician, salesperson and farmer. 

The participants were recruited in different ways during the years 2011–2020. A total of 105 of the informants were recruited through a snowball selection at organisations and enterprises. The HR managers at the organisations and enterprises were informed telephonically by the researcher and asked whether the organisation or enterprise was interested in participating in the study. The HR manager in turn directed the request for participation to the management in the organisation and asked for volunteers at different levels in the organisation who were interested in participating in the study. However, the sample of informants was voluntarily collected after the researchers invited the respective workplace to participate in the study and the organisation and the enterprise had accepted. A letter with information about the study was distributed through the contacts at the organisations and enterprises. The contact person informed people with the desired profile of participation and disseminated the information letter. The potential participants then gave their consent to participate or rejected participation. Additionally, 61 participants from an earlier work environment study conducted within the research group, were contacted by mail and received written information about this study with an invitation to participate in the study and to be interviewed. Twenty-two individuals responded and received further information on the study, were asked to participate and accepted and gave consent to participate. Twelve participants from an ongoing intervention study with managers were asked to participate and received further information on the study by phone and were asked to participate. Of those all accepted and gave consent to participate in the interview and in the study. The participants from the trade unions were invited in another way. The researcher e-mailed written information about the study directly to the trade union with an announcement and invitation for interested participants to volunteer for the study. The contact person informed people and disseminated the announcement and the invitation to participate in the study. The sample of voluntary potential participants was identified and subsequently gave their informed consent to participate in the interview and in the study.

### 2.2. Data Collection

The data were collected by focus group interviews and individual interviews during the years 2011–2020, using semi-structured interview guides with the same basic probing questions to invite the respondents to describe, based on their situation, e.g., what contributed to or would increase the possibility for them or their employees to be able and willing to work until an older age. Some of the collected data has previously been analysed for other purposes, i.e., why some had left working life early and others worked until older age, work motivation, the attitude between managers and employees, and the transfer of knowledge between generations [15,16,17,18,19]. 

Most of the data were collected through focus group interviews with 111 informants. Focus groups are suitable for obtaining knowledge about perceptions and beliefs regarding a specific topic [20,21]. The purpose of a focus group interview was to stimulate and start a dialogue between the participants to an open discussion within the scope of the study topics [22]. The focus group sessions included 2–7 participants from the same organisation or enterprise. The focus groups consisted of individuals in the distinct positions of first line managers and HR personnel; senior employees (55–72 years of age); and trade unions (Table 1). Two of the trade union workers (one man working at a construction enterprise and one woman working at a health care organisation) were also safety representatives, with the mandate of stopping work immediately if any risk or safety issues occurred in the workplace. The interviews were carried out in neutral and calm surroundings at the respective workplaces. Each interview session lasted 1½–2½ h and all the discussions were audio recorded and transcribed, although the identities of the interviewees were kept anonymous. However, due to time schedule and logistic issues, the interviews with 34 of the informants were individual interviews. The latter were intended to provide a deeper understanding and knowledge about the individuals’ own subjective experience, attitude and perspective of their situation [22]. Twenty-two of the individual interviews took place in the informants’ own homes, more specifically at the dinner table in their kitchens. Twelve individual interviews were telephone interviews. Each interview session lasted 1½–2 h and all the discussions were audio recorded and transcribed, but the identities of the interviewees were kept anonymous.

### 2.3. Analysis

All the interviews were analysed together through a text analysis method, deductive content analysis, to crystallise the relevant parts of the collected data. Deductive content analysis is a suitable choice when an existing theory involves the application of conceptual categories in the analysis of a new context [23]. Deduction can be said to constitute a conclusion from the general to the individual. The deductive content analysis increases the deductive approach by using theories and knowledge from previous research to refine, and possibly extend, a theoretical framework [24]. Additionally, in the content analysis any text that does not fit in the existing theory or the pre-defined categories is assigned new codes and is analysed to verify any new category; hence content analysis also follows an inductive approach. 

The analysis in this study was performed in several steps. The analysis started by constructing a formative categorisation matrix with four pre-set categories based on theories and the determinant spheres in the swAge-model [5,6,7]. In the next step of the analysis, all the interviews were read together as one text to make sense of the whole. So as not to miss anything of importance this was done twice. In the third step of the analysis, specific interesting parts were marked in the text and colour-coded. In the fourth step of the analysis, those colour-coded text parts were put together and given codes. The codes were then grouped and categorised according to their meanings, similarities and differences and linked to the pre-determined categorisations in different themes and sub-themes showing the reappeared basic ideas typical of the participants’ descriptions. The presentation of the findings in the following results section of this paper is based on the four pre-determined themes based on the determinant spheres of the theoretical swAge-model [5,6,7]. Some suggestions of measure activities are related to and supposed to solve issues associated to more than one theme context, to handle this they were sorted into more than one of the four main themes in the end.

## 3. Results

The result of measures and strategies to highlight in the organisational work, to make improvements for a sustainable extended working life for all ages, are presented under the following themes: Measures for the work environments health effects; Measures for personal financial security; Measures for relation, social support and inclusion; Measures for execution of work tasks.

### 3.1. Measures for the Work Environments Health Effects

#### 3.1.1. Measures to Consider the Effect of Biological Ageing Related to Employees’ Health and Risk Assessments in the Systematic Work Environment Management

The interviewees representing the various occupations participating in this study stated that the areas of importance in terms of decreasing the possibility of an extended working life were physical and mental health problems. The interviewees were of the opinion that a high level of physical and mental demands for many years in a problematic work environment often led to physical health problems, especially in the last year of employment. One participant from HR stated: *“I think that you can perceive a big difference after the age of 60. You lose very much then. Mostly physically. That you cannot be bothered, you are tired. It is hard to work in health care”.* A trade union representative from the construction industry described the health problems associated with increasing age: *“There are many employees who can take two painkillers, both at lunch and breakfast. However, most people go into the bathroom and take them, since most people do not want to show it. I dare not even speculate how large the use of painkillers is. It can range from 50% to 10%. If I must be honest it is very extensive”.* Many interviewees stated the importance of an action plan for organisational measures to examine work environmental risks, take actions and follow up on the measures and actions. A manager stated: *“We need to work with accident prevention all the time, such as to always wear protective goggles, gloves and the like. Such things have of course been an improvement to minor accidents today. But then the second thing, the long process where you work every day in a physically demanding occupation and eventually you will become physically worn out in old age. We have not really come up with how to solve this”.* However, there are actually systems on how to continuously manage these work environmental problems in the day-to-day work. One manager stated: *“If it makes a real impact to work with systematic work environment management then it is possible that more people can continue to work until an older age, I’d say”.* Interviewees stated that it is important to systematically work towards making the physical and mental work environments healthier and more sustainable for employees of all ages.

#### 3.1.2. An Organisational Culture That Promotes the Use of Ergonomic Aids

Many of the senior employees in physically demanding work environments described it as problematic to keep working due to the physical work environment. The organisational culture regarding the use of ergonomically correct positions, aids and equipment should be encouraged with the aim of improving physical work health. One manager stated: *“Younger ones just go for it. They hear well, see well and feel good. But when they are around 60 they have tinnitus for not using hearing protections when they were younger. Back pain all the time because when you were 25 you could lift 60 kg, no problem. The back creaked a bit, but it went well. Then it is too late”.* One trade union worker said: *“We have this macho culture: Just go for it!! I have been using the machine every day, year in and year out for ten years and it went well. I cannot feel my arms today, but it does not matter, just go for it. Then the younger guys think: If the senior employees work like that, I should not be weaker. Then they learn the wrong way and the problem continues”.* A manager in health care stated that it was obvious that the use of ergonomic aids can protect and enable the patients, but sometimes the employees were not diligent about protecting themselves and would take their own safety and health for granted. Another manager stated: *“It is important to inform more regularly about accessibility and protection equipment. That it is included as a requirement. We often focus on the patient, when the patient needs an aid we help them so that they understand how to use it. But it is also about our own working environment and health”.* The interviewees stated that it was important to instil a positive attitude to provide a good and safe work environment in the entire organisation. Furthermore, it is not only the managers, HR personnel and trade union representatives who need to be aware of this, the employees must also be educated and take responsibility regarding their own safety so as to enjoy a good work environment. 

#### 3.1.3. Rotation, Variation and Change of Duties to Reduce Physical and Mental Demands

Rotation between different work tasks was stated by the interviewees as a measure to reduce health problems from demanding work tasks and situations at work. One senior employee said: *“I am in favour of rotation between different tasks, and I talk from my own experience. You lay slabs and lift the stone, every day. You might lift 7–8 tons a day with your body. It takes a devilish toll on the body, this monotony. Instead, it is better to rotate a little and switch tasks within the work team. It’s about planning, to ensure that the conditions to be able to rotate exist”.* One manager said: *“You need to modify the task. When you come up to 60 years, you can’t roof a house, you can’t make roof trusses. Then there are other tasks. One must plan and structure the workplace. Can we put him in charge of small tasks, or shall we let him handle the logistics?”* To rotate could on the other hand increase fear, insecurity and the experience of more stress, if the senior employee was not used to rotate work tasks but only worked with the same tasks and in the same place. One manager described: *“When I started to work here I introduced the need to rotate on all workstations. Then there were older employees who had difficulty opening the computer at the new table, even though it was exactly the same kind of computer that also looked exactly the same. It was not possible to understand how to do it when they left the table where they had stood for 20 years. Some could not even enter their username. However, after a while it was no problem”.* To rotate work tasks in the workplace was also described as both a barrier and a solution to reduce problems in the mental work environment. It could in some aspects be perceived as stressful to move from a familiar work spot and work tasks to an unknown area, but on the other hand it could be a solution and a possibility to reduce stress and increase the understanding of the total organisation and towards each others’ work tasks. A senior employee stated: *“It would have been good to start from the beginning and to really get around. Many of the staff have to move. They have been in one place for 30–40 years. They do not know what it looks like at their neighbouring colleague basically. It is very good to walk around and see what others have it like to appreciate their own, and maybe come back again”.* In other words, a regular rotation between different work tasks seems to be a good measure to promote a healthy and sustainable working life for all ages.

#### 3.1.4. Communication, Information and Participation to Reduce Work Stress

The interviewees described a problematic work situation as when there was a great demand on them to execute their work tasks, even though there were a lot of factors they could not control in their work situation, affecting their ability to execute their work tasks. Additionally, there was organisational development underway that the employees did not understand or perceive as having any possible objective or benefits based on their own position within the organisation. A trade union representative said: *“There are so many reorganisations and we face a lot of new systems. Our members become much stressed if they do not understand why there must be a change or because of the novelty of it. You become frightened and tired of all the new systems […] it could be new and changed monitoring systems, payroll systems, personnel systems, financial systems, planning systems. We need to receive information and be included in what and why these systems are needed. If they are needed”.* The interviewees described that better sharing of information and worker participation in organisational development and changes in work tasks could increase their understanding of what was going on within the organisation. Therefore, they suggested more accurately targeted information about, and participation in, organisational development and work task changes as important measures that could balance the sense of reward and decrease work stress. The senior employees also described how they as employees want to experience a sense of reward for the effort they put into their work tasks. The older employees described how work tasks would not bring fulfilment due to circumstances beyond the employees’ control, and that words and actions of appreciation from their managers and organisations could improve their situation, by helping them to better manage their experiences at work and reduce their stress levels.

#### 3.1.5. Reduce Violence and Threats in the Work Situation and Brief Each Other in Support Groups

There are times in the work situation circumstances that can include threats and violence. A senior employee in a health care organisation described: *“There are some patients who are confused and demented and who can become violent and fight. There are also patients with frontal lobe dementia who become completely personality-changed and pull staff down in bed and try to make sexual encounters. They can’t help it, but it is our work environment and it is stressful to face this every day and having to defend ourselves”.* Undesirable situations that include violence and threats must be managed properly and reduced so that a higher number of employees of all ages can keep working. An important measure described by the informants was to hold briefing sessions regularly with each other to talk about possible, perceived and experienced situations and issues in the work situation including threats and violence. The possibility of talking about these situations in support groups and with supervisors was described as a much-needed measure to reduce stress and anxiety regarding new situations including threats and violence and on how to handle them. 

#### 3.1.6. Work Schedules

Measures regarding the working hours appeared to be important to make working life more sustainable for all ages. An HR practitioner said: *“Many senior employees like to reduce the number of work hours, have shorter work shifts and more flexible working hours to cope and have time to recover”.* Flexibility in the work schedule was stated as important for senior employees’ revitalisation and to be able to work until an older age. A manager defined it: *“The possibility of having a little bit more flexible working hours is one of the most important factors for the possibility to keep working in an extended working life, as I perceive it”.* However, many stated that flexibility in working hours was not feasible in many organisations since it would impact the production line. In contrast, others stated that this was not true, because it could be done with better planning and organisation of the work tasks. Another HR practitioner said: *“Many senior people prefer to work fewer hours because they are unable to cope. I think younger people would rather work a very long shift and then have time off for two days in the middle of the week. When you are older, you have changed that view and gained the insight to work more often but shorter work shifts to cope. We especially saw this in the workplaces where there was an opportunity to influence working hours and schedule. There you could see that younger people worked much longer work shifts but more rarely. Older people worked more often, though had shorter work shifts, preferably five-hour work shifts”.* A trade union worker in a construction enterprise stated that many senior employees no longer had the strength needed to work full time and said: *“Those who have come up to 60 years and have physical problems could work part-time, and go down to, say, 75%. If the senior employee had reduced their working hours, by working fewer hours every day, for example from 09:00 to 14:00 (5-h work days), it would not affect the work team and production negatively at all […] In more project-oriented workplaces, for example, the staff could work for three months and then have time off for three months”.* However, the trade union worker also mentioned that if senior employees decreased their working hours their pension could be negatively affected, and therefore many senior employees do not consider reduced working hours as a possible choice to have a more sustainable working life. 

#### 3.1.7. Work Pace

The senior employees were described to be just as exposed to the demanding mental work environmental problems as the younger ones, i.e., sometimes they experience an insufficient influence of their own decisions and control on their work situation, and some run the risk of being subjected to threats and violence. However, it seems to be especially stressful for the ageing employees with high expectations on their productivity from enterprise/organisation, managers and co-workers when they, due to their biological ageing and health problems, cannot be as productive as before, in some work tasks. A trade union worker described this: *“There are difficulties in the work team when older employees are worn out and cannot perform as well as the other ones. Not to be performing fully anymore and risking dragging down the contracted work in the work team is stressful for the older employees”.* The work pace was mentioned when it comes to improving measures for senior employees. One manager stated: *“I feel that those who are 50, 55+, they cannot work at the same work pace of those who are 25–30 years old”.* Additionally, many of the senior employees stated that they experienced that work was more stressful these days and that they could not work at such a stressful work pace. Instead, they wanted fewer work tasks; to have the possibility of focusing on fewer tasks, but performing better. Another important measure described by some interviewees was to adapt the work content to the scope of working hours, so that those who work part-time were not expected to perform full-time work in fewer hours. Too many work tasks increase the work pace because it also takes time to switch between different work tasks. A senior employee in elderly care said: *“There are so many work tasks around that have nothing to do with care work at all, and that is stressful. We must document a lot, write a lot and keep contact with relatives. They pull at you from all directions. We must bake, we must get food ready, we must organise activities to the accommodation, and we must pack up diapers. We have to do so many different things, the patients have to wait. The patients complain about that, and then you get yelled at by the manager for not having time to do the care work”.* The interviewees stated the importance of good quality in the execution of work tasks, and not to focus as much on quantity. The interviewees described that a measure to achieve this was to have fewer tasks and work at a more comfortable work pace where they could be more considerate and creative in executing their work.

#### 3.1.8. Importance of Self-Care for a Sound (Occupational) Health

The interviewees stated that healthy ageing is about taking care of themselves and their own health. However, it was stated that this was easier with support from the work situation and the organisation. Some interviewees talked about the fact that individual employees have a responsibility of taking care of and managing their own health, but that the prevailing attitude, for example in the workplace, contributed to whether they could and whether they took this responsibility. A manager said: *“There must be a genuine interest in taking care of oneself, because it is not just about exercising, it is also about eating right and sleeping properly and all those other things that affect how we feel in everyday life”.* A senior employee said: *“To get exercise, eat, sleep, and all that influences how we feel in our daily lives is also a part of the working environment”.* Some interviewees spoke of the significance of physical exercise, but also the importance of a healthy diet. The interviewees also stated that people had better eating habits before, when it was possible to get coupons from the enterprise or organisation to exchange for lunch. These days this counts as an income benefit in Sweden and results in increased taxes for the employees, therefore no enterprise or organisation has this system anymore. However, there was a greater concern about the younger generations’ possibility of having a sound health. The managers and senior employees pointed out that the younger employees preferred eating fast food instead of maintaining a healthy diet. A manager said: *“We have many people who have a Coca Cola and a chocolate bun for breakfast. I think it would be a good measure to treat the staff to a good breakfast when they come here, so they have the energy to work”.* The manager stated that the young employees of today will be elderly employees one day, and that if they do not change their diet, they will have serious health problems when they get older. The interviewees discussed that physical activity was very important to keep mentally and physically healthy. An occupational health care professional said: *“Fitness in some way is essential for everyone to maintain a sound physical and mental health regardless of desk work or physically demanding work. Exercise either at work or sponsored by work. Exercise at work could be mandatory, not only for police officers and firefighters as it contributes to higher productivity and less sick absence among employees”.* A manager said: *“I don’t think there’s a chance that you will be able to work until an older age unless you take care of yourself and your body. It can well be stated that a lot of our employees have physically demanding jobs in health care, but who also miscalculate their body if you look at how they maintain their physical exercise and their diet. It’s not good. We do what we can there. We provide information, offer lunch and occupational health care and tell about the importance of physical exercise, mental recuperation and a well-balanced diet. That you should exercise even if you have a physically demanding occupation”.* Some interviewees discussed whether measures to develop an organisational culture that promotes self-care and a healthy lifestyle could be a step on the way to a more healthy and sustainable working life, or if it was to violate employee integrity and risk of shaming people.

#### 3.1.9. Physical Activity, ‘Maintenance of Functions’, to Sustain and Improve Mental and Physical Good Health

Physical activity was stated as an important measure to sustain and improve sound health, both mentally and physically. Organisational support for making it possible and, in addition, compulsory, to exercise at work and as a part of the work schedule, was a measure supposed to support a more sustainable working life until an older age. An HR practitioner stated: *“I come from the police department and also work quite a lot with firefighters. In those occupations physical exercise and maintenance of their body during working hours is mandatory, to score their tests and keep up good physical form in order to do a good job. But it’s almost more tiring and demanding to work in health care. Health care professionals need to exercise as much as employees in the police department or fire brigade. It should be equivalent when you save people’s lives. I think that all physically demanding service occupations need to get exercise at work, as part of the working hours. It should be a requirement, because it’s a safety and work environment issue”.* A senior employee said: *“I think it would be good if they would put fitness exercise in the work schedule. I think many do not want to go away when they get home. They live far from the city where they work. But if they had one hour a week in their working hours I think, if you put it in, many would feel a lot better because of it, it is actually perceivable. We see it in our elderly patients’ exercise and raising of their arms every now and then. Just doing that a couple of times a week makes them feel better. Why can’t the staff do that, without having to do it in their own leisure time? We get older, as do our shoulders and arms, our bodies shrink, it’s really important to exercise. I think most would agree if it was in the working schedule. Don’t you too? Now that us co-workers exercise for a few hours, and gently move our arms and shoulders, nothing unusual really, so many have felt good because of it”.*

Another senior employee described that physical activity was not only important for the body but also for the mind. She said: *“You become more alert if you exercise at all, because I know some girls who have started and never done any before, and they say that they are very energetic: I can do more than I did before! And I actually think that you do”.* Physical activity and exercise were highlighted by the interviewees as an important measure to promote a healthy and sustainable working life for all ages. 

#### 3.1.10. Occupational Health Services Support to Prevent Work Environment Problems and Increase Good Occupational Health

The interviewees stated the need for measures to make working life healthier for all ages and to make working life more sustainable. Some managers and HR practitioners emphasised the need to place the employees’ wellbeing in their systematic work environmental management to handle an extended working life. Someone said: *“When we do our safety rounds and look over tools, activities, etc. we can also look over the employees’ physical and mental capacity to work”.* However, the HR practitioners and managers further stated the need for professional help to decide on and implement the measures. An HR practitioner said: *“I think we need to take more action when I see the injuries that employees have today. This will be intensified if we do not begin to take measures in the working environment and work situation now. Measures must be reasonable in some way, if we expect that elderly employees will perform and deliver until an older age in the same way. Perhaps measures for physiotherapeutic rehabilitation. It is not only the somatic part. It is so easy to decide on and take physical action. But it can also be mental. It should not be forgotten. This second part is more difficult to work with. We have occupational health services that we can turn to for help with it, but maybe we also should have something in the administration that we can turn to”.* As the HR practitioner described, the managers have many tasks and responsibilities and they cannot be experts in every field. Therefore, they need to include other professionals and experts to take care of some of the measures in order to make the work situation healthy and sustainable for employees belonging to different age groups.

#### 3.1.11. Summary about Work Environments Health Effects

The participating interviewees stated that an awareness of employees’ ageing in their systematic work environment control management and strategy in the daily work at the organisation or enterprise was an important strategy for sustainable working to an older age, i.e., an awareness of ageing in the investigation of working conditions; in the assessing of risks; in the development of an action plan, and when to take action; as well as an awareness of ageing in the follow up of the results. Other measures to promote a good and healthy work environment that supports health and wellbeing, and are of importance for a sustainable working life until an older age, were: an organisational culture that promotes the use of ergonomic aids and tools; rotation and change of work tasks to reduce physical wear; physical activity to maintain bodily functions and keep mentally and physically fit; the importance of a healthy diet for sound (occupational) health; and occupational health care support to promote health and prevent physical and mental injury, illness and stress. 

### 3.2. Measures for Personal Financial Security

#### 3.2.1. Salary and Financial Benefits

One reason to work is to receive salary to finance one´s life. When we asked the senior employees about their reasons to keep working in an extended working life some stated: *“For me it’s a financial issue. I am not hypocritical about that”.* A trade union representative said: *“When you work you have reasonably good finances. But, on the other hand, you have less leisure time to do activities than someone in retirement. As a pensioner you will have to live with less money, but much more time to do what you want. It’s a balance. When can I leave (for retirement)?”* The changes in salary and financial benefits are measures that can work both as a carrot and stick for a longer working life.

#### 3.2.2. Measures in the Organisations and Enterprises Work Environment to Promote Continuous Employability 

Some interviewees distinguish the need for measures to promote continuous employability, so that senior employees can provide for themselves through continuing to work until an older age. A necessity is that the employee’s health is sufficient to work, to be able to participate in working life and to receive a salary. A manager stated: *“Elderly people who are worn out and have a sore neck and shoulders, they get a medical certificate that they cannot lift more than 10 kg. But then we say to that person that we do not have any tasks for them. Because we have not any work tasks where you do not have to lift at least 10 kg. So then there is no possibility of re-employment after their injury and sick absence”.* The interviewees stated that work environment security to reduce risks of occupational diseases, injury, sick absence and disability pension were important for the individual employee’s employability and personal finances. A manager said: *“20 years ago, people were unable to work until retirement age in this profession. Because back then you did not have the tools and you did not have the working environment of today. But even today, some are worn out but have to suck it up and work because they have no choice. Otherwise they will have too low pensions and have financial problems”.* Some interviewees stated that the organisations and enterprises were responsible for the working environment, that when the organisation or enterprise chooses not to provide safety aids or tools or does not provide enough people for an activity, they should consider the long-term cost of this since it can cause sickness absence and disability benefits in the long run. A trade union representative said: *“In large workplaces, it is easier to provide more people and protection equipment than in a place where we are two employees, because the price is so bad, we have not considered the cost of that. So, in those workplaces you have to work harder than one would have thought. We also have some smaller workplaces in facility work where we do not have access to the work protection equipment one would need at all times. It is often like that. What to say of the rules of preparation, it is usually included in larger projects, you need this and that, and then it is part of the offer and price, etc. That is where the safety representative should object”.* The interviewees regarded work environment safety representatives at the workplaces and their mandate to stop work if there are any health risks present as an important measure to reduce risks in the workplace that could affect the employees’ health, continuous employability and by extension, employee salary and personal finances. Some interviewees, mostly managers, highlighted the importance of safety protection, and how injury and sick absence can cause economic stress for the organisation or enterprise. 

#### 3.2.3. Measures to Change the Organisational Culture, to Promote and Increase Responsibility and Employees’ Use of Safety Equipment and Assistive Technology in the Workplace

The organisational culture influences how one perceives one’s work, tasks, as well as the use of aids, tools etc. Therefore, the organisational culture should promote a responsible and sustainable working life, to prevent employees being worn out physically or mentally. It is also important for the staff that the management shows appreciation for their efforts. A trade union representative stated the importance of taking measures to change the organisational culture: *“We have this macho culture; just go for it! I have used that machine every day for years and years, ten years and it went well. However, I have numbness in my arms today, though that does not matter, just go for it! Now I generalise a little. When the young guys learn, the frail Bengtsson can do it, he is able to, and so am I! Then, for example, we have a young guy today, he is not yet 30 years of age, who must have cortisone pumped into his arms. Not yet 30 years old! It is too weird that it should be like that […] they are 30–35 years old and are starting to have physical problems with knees, back and legs already, and they may not have worked for more than 15 years. After all, it’s scary that many of the younger ones can’t handle it”.* Many interviewees working in physically demanding workplaces expressed the need to change the organisational culture, to use the aids and tools that were available to avoid being exhausted and unable to work, and subsequently risk financial consequences due to lack of employability. A senior employee who expressed concerns about the organisational culture said: *“Younger people go on just like anyone else. You hear well, look good and feel good. But around 60 you get tinnitus for not using hearing protection, so it is, sore back because when you were 25 you lifted 60 kilos no problem, it creaked a little, but it went well. That’s what it is like. Then it’s too late”.* Opportunities to maintain employability until an older age and to counteract “macho attitudes” were described as conscious efforts aimed at creating a more favourable organisational culture. The role of the organisational culture in employability at an older age was expressed, above all, by managers in the heavy construction industry. A manager stated: *“**There’s a huge focus on working environment, actually. […] It was stated by the executive group management that it is not about money, it is about wellbeing at the employees construction site”.* A manager stated that acute injuries were reduced with protection equipment, but that becoming physically worn out seems more difficult to combat using physical means: *“Now we have introduced to always use protection goggles, gloves and the like. Such things have been an improvement to minor accidents today of course. But then in the long run, when you work in a physically demanding profession, you will get worn out eventually”.* Rotation between different work tasks was one measure stated by the informants to reduce health problems in a working environment with heavy physical demands.

#### 3.2.4. Measures to Promote and Increase Employability through Continuous Competence Development

To execute work tasks and activities the employees need appropriate knowledge and competence. Many of the interviewees stated the need for continuous knowledge and competence development for the employees to stay employable. However, some managers, and some senior employees, stated that some of the senior employees do not want to continue developing their knowledge and learn new things. Most organisations and enterprises must undergo continuous development due to the ongoing change in the world and societal circumstances. If an employee does not have the right competence and know-how required for their work tasks, they are not employable. A manager stated: *“When there is change in the organisation, which of course it happens for a bit every now and then, if the employee’s competence does not fit the new tasks, they should be placed somewhere else. Then we try to get these employees other tasks and locations. We also need to look at what they bring in competence. Sometimes a senior employee no longer has the right skills and training. They are not as employable anymore”.* Other interviewees stated that with a broad competence and/or special competence it was easier for the employee to change work tasks, workplace or occupation if health issues forced them to do so. The participating interviewees stated that it was of great importance to take measures to promote a continuous improvement of competence, personal development and learning of new skills in order to maintain employees’ continuous employability, and in the long term, personal financial security though the possibility to obtain salary from work.

#### 3.2.5. Summary about Personal Financial Security

The interviewees stated that measures to promote personal financial security for senior employees were important to secure their continuous employability. Therefore, measures to ensure work environment security, risk assessment and reduction of work injuries are needed to reduce the risks of health impacts, so that employees can manage an entire working life and not be worn out prematurely and subsequently forced out of working life with less financial benefits through long-term sick leave, disability pension, unemployment or premature retirement. Furthermore, continuous competence development is key, so that senior employees do not risk being laid off due to lack of fitting tasks to workers because their competence is obsolete, or they lack the skills needed and therefore cannot be relocated with continued employability and provide for themselves by participating in working life. Some interviewees also stated the need for a reasonable and sufficient salary regardless of age. 

### 3.3. Measures to Promote Relations, Social Support and Inclusion

#### 3.3.1. Measures to Promote Work–Life Participation by Balancing Working Life and Leisure Time

Employees have a personal life outside their work. The interviewees describe a need for measures in the work situation that utilise individual needs, participation and activities with family, leisure time and hobbies. Many of the senior employees stated that they were much more tired these days. This fatigue affected their leisure time, because they had to prioritise rest and recuperation when they did not work. A senior employee said: *“In the evening when I came home, I did not have the energy to start doing fun things […] I have way too little recuperation. I got a telling-off from my brother last summer; he said that I never call and never come to visit. I replied that I do not have the energy. After work I go home, cook, eat, sit on the couch and then I go to bed. /… / Over the years, I feel that the balance has got worse and worse […] We have a little cabin in Falkenberg where we would like to spend some time… But we can’t, because I don’t have time… I don’t have time to do anything at home”.* Being too exhausted to take part in leisure activities and exercise could cause feelings of embarrassment. A senior employee said: *“It is like you said previously, I am so tired. I am almost ashamed sometimes that I don’t have the energy to do anything, anything, other than work and sit in front of the TV. It is like that unless I am off for the whole weekend or something. I don’t really do anything during the weeks other than work. It almost feels a little bit embarrassing”.* Many of the interviewed senior employees stated the need for measures to promote senior employees’ work schedules and working hours that, other than work, address the need for the individual’s participation in a personal social environment, leisure activities, family, hobbies and relaxation. Many state that they would prefer to leave working life due to the lack of content in life besides work, and to have a better possibility of living a full life.

#### 3.3.2. Measures to Promote Social Inclusion in a Team to Increase Participation and Community

The sense of community in the workplace was described as important to create a sustainable extended working life. Difficulties arose if the senior employees sensed feeling like outsiders in their former work team, in which case they would not want to extend their working lives. A trade union worker said: *“There are no guidelines in the enterprise generally saying that when a skilled employee, coming up to 63, 64 years of age, then it should be done like this. They leave it to the team to redistribute responsibilities. There will be an outcry: we should not have him in our work team, because he is worn out! He cannot perform piece work! Actually, piece work is an issue for senior employees”.* All the interviewees stated the importance of the work social environment and measures to promote inclusion of every individual in a reliable team. To be a part of a social group was stated as important to increase the willingness to keep working in an extended working life. An HR practitioner stated: *“Those who appreciate the social life in their work situation and who have a large social network at work tend to work for quite some time”.* A senior employee said: *“I get on well at this job and it feels good. I meet many nice colleagues and managers who appreciate the effort, so there is no hard consideration to keep working”.*


Another senior employee said: *“One thing that makes you want to come here and work is that you have a lot of friends and acquaintances. People you meet every day, talk and hang out with”.* An organisational culture with sustainable values allows customisation of the work situation. The sense of participating in something larger than oneself and working with colleagues were stated as important factors to keep working in an extended working life. 

#### 3.3.3. Measures to Increase the Senior Employees’ Status in an Occupation and a Work Team

The interviewees describe that it is key to feel that work tasks and activities are perceived as important and needed for the productivity and the organisation. Having appropriate resources and equipment to execute the work tasks was seen as important in order to experience appreciation from the organisation and the manager. Some production and professional groups included in the study had received new work uniforms and were offered training days, which assisted in raising the status and value of the employees to the organisation. A senior employee describes: *“It was Anna (a manager) who introduced it. She cared a lot about raising the status of their careers. So they got uniforms that say ‘Cleaning Department’ on them, and they all got training. A whole week of training. It cost a lot of money, but I think it made them feel more appreciated”.* However, the participants described that male-dominated occupations and work tasks have a higher status within the organisation and are more appreciated than female dominated occupations. A participant stated: *“Men have higher status in organisations. All who needed got work uniforms that were paid for by the organisation, and they do not have to take them home to wash themselves. They have full time work, and also have a higher salary for the same work tasks”.* Some participants implied that this difference in status was part of the fact that more women left working life earlier than men. They stated it as an important measure to promote and increase the status of female-dominated occupations and work tasks to increase their willingness to remain in an extended working life.

#### 3.3.4. Social Support to Promote and Increase the Senior Employees’ Self-Esteem in the Organisation

Social support was described as a measure to increase the older employees’ willingness to stay in working life for longer. One manager stated that many senior employees had to change their own self-image by themselves too, and said: *“It is often the general idea in society that individuals are not interested in the labour market when they are over 55. This is in many people’s heads. If you do not believe in yourself then it will be tough”.* Another manager said: *“I have employees who basically have the same chronological age, but where one has decided to work until 67 and the other to work until 65. I think many start to dip down when they begin to see the end of working life in any way. Then, it is a lot about motivation, and I discovered that it determines quite a lot and that’s pretty much about it. Attitude and motivation affect a lot”.* Some interviewees stated that because of old age, health problems or personality, some employees do not want to be in the front line. They want to take it easy, have enough time to recover and do a good job at their own pace. Therefore, a means to keep a larger number of senior employees within the organisation until an older age was to make it possible to have different positions based on the senior employees’ own needs. However, they need to have the ability to know that they are good enough and to feel included in the social participation despite not being able to keep up with a fast work pace. A manager said that they have made agreements within the work team to make it possible for everyone to get best, or at least good enough, fitted tasks: *“There was an agreement and this elderly nurse has since then expressed that she feels very calmed by it. She does a satisfactory job, but she does not need to do this to be in the front line. […] She avoided the stress, she has expressed that; now I know what I should do, I will do what I must. For it is not fair that one should leave a long working life with a sense of failure; I’m not good enough. Then it is better to have taken that into consideration”.* Measures to promote and increase an employee’s self-esteem in the organisation, irrespective of the employee being the most productive worker or not, were said to be important in remaining in an extended working life.

#### 3.3.5. Measures to Promote and Increase the Attitude of Employees as a Productivity Investment

The interviewees stated that it was an important measure to feel included in a social group and to be seen as a unique individual to make work–life more sustainable until an older age. 

All the respondents stated the importance of different age groups at the workplace because it takes time to build experience from life. One of the trade union workers said: *“It takes such a long time to build up experience that some older key people are worth gold to the organisation”.* Senior employees were described as a resource offering valuable assets for organisations and companies. Measures to take care of and promote the senior employees’ experience-based knowledge were needed both for the acknowledgement of the senior employees and for the prosperity of the organisation. Some managers described that they included senior employees in development meetings for the business and new projects even though the employees were older and did not have much time left before retirement. A manager stated: *“It is very valuable to have a senior employee in the production and in the work team, they know what to do and have been through most problems so they keep calm in the most difficult and problematic situations”.* Having mixed age groups strengthens creativity, growth and flourishing because experience and knowledge meet and can be exchanged. To highlight and promote the senior employees’ value to the work team and the production was described as an important measure in the quest for creating a healthy and sustainable extended working life. Some of the respondents also described how customers and patients who themselves were elderly preferred to turn to senior employees with their requests. This is probably because they perceived it to be easier to meet in a common reference framework and historic familiarity with someone in the same age group that facilitates the communication. 

#### 3.3.6. Measures and Actions to Decrease Negative Attitudes and (Age) Discrimination

Managers, HR practitioners, trade union representatives and senior employees stated that it was important not to generalise, not to hold stereotypes that all senior employees are the same and take actions and measures to eliminate negative attitudes toward ageing, victimisation and age discrimination because of effects on work ability related to biological and cognitive ageing. The attitude towards senior employees held by managers, co-workers and organisations influences the senior employees’ experience of motivation to work. However, in the analysis it was possible to determine some negative attitudes towards senior employees in the work organisation. A manager stated: *“To put it bluntly: it is a fact that you want to invest in the younger employees, they’ve got many years left. Now we have laws and regulations that govern us and control us to not make any difference because of age”.* Another manager described how older employees’ productivity was not as high as the younger employees’: *“I may, as supervisor, not expect the same productivity of a 65-year-old employee that I can of a 30-year-old. But is it okay that older employees are not as effective? Is it okay if they cannot do the same things? That they cannot produce at the same level? Can we justify that we expect different things at different ages from our employees? For it is like that in reality”.* Some senior employees indicated that they felt discriminated against because of their age, and that it was important to change this perception. One senior employee said: *“I heard at some point that when you as an older employee continue and work for longer, you do not make place for younger generations. I do not know how to understand this because it is not really true. It is not a matter of a generational switch. It is about competence. You should see us more as individual employees with different competences, instead of a certain age”.* The participants stated that it was important to eliminate negative age-related attitudes and to acknowledge individuals instead of generalising about individual employees based on their chronological age and on stereotypically negative attributed characteristics of the social age group elderly employee, i.e., to age discriminate. 

#### 3.3.7. Summary about Relations, Social Support and Inclusion

The interviewees stated that measures to promote social inclusion, participation, coherence and social support in the work situation by considering whether all employees were included in the work social environment and in the work team are of great importance. They also stated the importance of measures to decrease negative attitudes and (age) discrimination, and to increase social support and the senior employees’ self-esteem in the organisation. As well as to appreciate the senior employees’ mentoring, (working) life experience and calming effect on the work team as an important productivity investment. Additionally, employees have a personal life outside work and senior employees, due to their biological ageing, need more time for recuperation, the work schedules need to pay attention to individual needs for social participation outside work and activities with family, leisure time and hobbies.

### 3.4. Measures for Execution of Work Tasks

#### 3.4.1. Measures to Increase Motivation and Work Satisfaction

At work people must perform work tasks and activities to receive their salaries. Many interviewees describe that their work tasks and the content of their work activities were very important to them. This was especially the case with older employees with work tasks and activities involving problem-solving, where they could utilise their abilities and skills in a way that they could not do outside of work, who stated that they did not want to retire from working and that it was of great importance for them to stay in working life. However, other older employees stated that their work tasks and activities lacked meaning and were a reason for them to leave working life. One senior employee stated the need for measures in working life to make the work interesting, motivating, meaningful and stimulating, the employee put it like this: *“If you go to the same place and do the same thing for 40 years, then it is not as much fun. We receive new challenges when we get into new projects, with new people, and in making sure that it works”.* To sometimes declare and highlight the employees’ work roles, tasks and activities in the bigger picture of the workplace, to the production and in society, was described as a measure that would increase the experience of the tasks and work activities as motivating, meaningful and appreciated within the organisation, enterprise and society. Measures to promote the experience of work tasks as interesting, meaningful and stimulating, or the experience of activities together with co-workers as stimulating and meaningful, were described as important in order to keep working until an older age. The feeling of importance when performing work tasks was described as meaningful and stimulating by one senior employee older than 65 years, who stated: *“I still feel curious, I am not fed up by what I am doing at work, but find stimulation in it all the time”. Another senior employee said: “I would not have kept working if I had not been stimulated. It applies to conditions and everything”.* Many participants stated that the experience of importance in work tasks constituted a preference to remain in an extended working life. 

#### 3.4.2. The Rotation and Change of Work Tasks in Order to Increase Motivation and Work Satisfaction

Some interviewees described the rotation between different work tasks to be a measure that motivates and stimulates employees to remain in an extended working life. One manager stated: *“I think we need to have rotation. It should be mandatory to work in different places and to move around at work. Between different tasks, within their own workplace or in another department. Because I have seen when we have forced people to move around, at first it is only disastrous for this person. They believe that working life is over. Six months later, when you ask them, they say it’s quite amazing, really good: "I have had to learn again, I have seen new things and met new people" they say. I have never heard anyone say that it was a disaster when it’s been a while. In the beginning of the change many are paralysed by fear, but then after a while it is only positive”.* One senior employee described how she got a new job after a reorganisation, including rotation between work tasks within the work team, she stated: *“I love the contacts, the meetings you have with new people, it doesn’t matter if they are older or younger. There are always new meetings, new challenges; how do I solve this? And then the wellbeing of the team. A bunch that always stand up for each other, you have problems and crises within the team that you always have, so you can sit down, nurses and manager and everything, so it is a great concept and great manager, it helps a lot. The manager, she trusts the work team. It is a security for us”.* However, some of the interviewees stated that to make rotation possible the employees had to have the skills needed for various work tasks. One manager said: *“Rotation is important. But the skills issue is very important at that. It’s about people who think that it might be good to go on and widen their views and their areas of expertise, otherwise it will not work. To participate in knowledge development, or to read up on their skills by themselves”.* This manager also described how it was his issue to motivate the employees to develop. Individual development and broader know-how from the rotation of work tasks was also described as an important measure for the employees’ employability and possibility to keep working in a changing work organisation and working life. However, the interviewees also stated that not all employees could rotate and do every work task in the workplace due to functional variation in physical or mental capabilities.

#### 3.4.3. Measures to Highlight the Employees’ Abilities

Some of the participants stated that in work and at the workplace, the individual employee sometimes becomes anonymous and assumes the role of an employee to perform the assigned tasks and activities at work. The individual’s unique abilities run the risk of becoming invisible and of not coming to fruition. To experience oneself as a replaceable cog in the organisation’s machinery was described as having a draining effect on the motivation. The meaning of their own individual efforts was experienced as non-existent by employees. Some of the senior interviewees who had left working life at an early age, i.e., before 64 years of age, stated that they had experienced work as a barrier to do more meaningful and satisfying things with their life and that they would have gladly stopped working even earlier if their personal finances had allowed it. One interviewee who felt that he did not receive any appreciation at work stated: *“I was quite skilled at finding problems in the energy system and solving these problems so that the organisation could save a lot of money. But, they never thanked me for that. I sold myself cheap at that job and never got any credit for my commitment”.* Another senior interviewee said: *“New managers, who were economists and did not know anything about the work tasks and how to do things, changed the organisation. You and your work team couldn’t decide on how to execute the work tasks anymore and there was much more stress”.* One interviewee made this statement: *“I put so much into that work but nobody cared and no-one appreciated me anymore. My manager frankly did not give a damn. It was no fun anymore […] if someone cares and appreciates what you do, you want to do it even better, but if no-one cares you stop caring too”.* This type of experience caused frustration and made work feel dull and uninteresting. Therefore, many of the participants stated the importance of paying attention to the employees’ individual skills and specialities and to highlight them as unique individuals, important to the functionality and productivity of the organisation or enterprise. Some managers described how they had made their employees responsible for different areas and tasks to increase the motivation and the employees’ experience of being needed and required by the organisation. Some managers also described how they had noticed some employees’ leisure interests and utilised those skills in new work tasks for the employee in the workplace, which had been a success for both the enterprise and to the employees’ motivation to work. 

#### 3.4.4. Measures to Promote Competence Development in Order to Enable Continued Employability

To execute their work tasks the employees need to have the appropriate knowledge, competence and skills. Some interviewees stated that competence development needs to be continuous to address the changes in work, tasks, as well as to meet changes in the world and technology development. An HR practitioner stated: *“As long as you work you need to have the right skills to do the job. That is pretty simple. The day you stop training is, well, on the day you go home”.* That the staff has the appropriate knowledge and abilities lies not least in the interests of the enterprise and the organisation. It is costly to lose competence and to have to recruit new employees. A manager said: *“Retaining and developing staff is important for the work and business. It is true that hiring a new mechanic, it costs about a million before they are up and running and fully productive. And if you let go of a mechanic who worked here, you must start all over again. We continuously train a huge amount here in the workplace as well. And we did not have to dismiss anyone for that reason”.* It was described as an important measure both to the employees and to the organisation that employees are enabled to continue their employability. 

#### 3.4.5. Competence Development Regardless of Age

Cognitive ageing affects individuals’ reactions, memory and ability to store knowledge, this was described by some of the interviewees. A manager said: *“Elderly may need more time to learn new things that are not in line with their previous knowledge. But I have not worked with someone who had to leave his post because he could not handle the new technology. But it’s just giving them different durations of time”.* Unfortunately, there were managers who indicated that they did not really see the benefit of training and developing the competence of senior employees who would be leaving working life soon. A manager said: *“To be completely honest, it is the younger employees that we want to invest in. The elderly are already on their way out of here”.* But there were also managers who, on the contrary, saw it as more important to invest new knowledge and competence development in the senior employees because they already had extensive knowledge, therefore they could add more value to the business directly. *“I can see it almost in the way that in younger employees you have to invest a lot of money. You must educate. You must make sure they go on. But then when people are over 50, then you can harvest. Then you get the return! You don’t have to keep them going. They know their stuff! They are self-sufficient! They take their own development initiatives to the extent needed! You get a lot of stuff. It’s harvest time!”* Many of the participants stated that an important measure to enable employees to continue working until an older age was that they continued to develop their competence and skills until the day they ended their working life.

#### 3.4.6. Organisational Culture That Acknowledges and Utilises (Senior) Employees’ Experience and Knowledge

On having lived a long life, a person accumulates many positive and negative experiences and generic skills that can be added to book learning from formal education. The use of this experience-based knowledge was highlighted by several different interviewees. The participants described how measures to enable the senior employee’s experience-based knowledge to be utilised in the work tasks were valuable for the senior employee, being able to use and get access to this experience-based knowledge, which also contributed to their experience of feeling valuable, and that this experiential knowledge was a valuable asset for organisations and enterprises. A senior employee said: *“It is important to utilise skills. Leave us the freedom to do our job based on our expertise. So that we can help and support if you need to discuss something, and at the same time that there is a possibility for us to have support if we need support at work”.* Measures to transfer and exchange knowledge and competence between the generations were identified as a significant investment to both new employees, the senior employees and to the work organisation. A manager said: *“It is important to have the opportunity of utilising the elderly’s competence and commitment. They may practise some sort of mentoring and transfer their knowledge to different teams”.* Furthermore, senior employees who possessed special skills were more interesting for the employer to retain and they gladly met these employees’ demands for adaptation of work tasks, just to be able to keep these employees until an older age. A trade union representative said: *“It takes a long time to build experience, so some people with what they have gone through are worth gold to the organisation./…/ I think it makes you a little bit special to the enterprise and you have better opportunity in negotiations. Maybe that you can work three days a week, then we make this deal. Then these people (senior employees) feel a little bit like, a little proud, you see”.* Measures to promote and increase an organisational culture where the employees’ knowledge, no matter if it is experience-based knowledge or knowledge from education, is utilised and considered important to the organisation, seem to motivate and stimulate senior employees to keep working until an older age according to several of the interviewees.

#### 3.4.7. Summary about the Execution of Work Tasks

Measures to promote knowledge, competence development, creativity and intrinsic motivation in the performance of work tasks were described by the interviewees as an important strategy to enable employees to participate in a sustainable working life until an older age. Furthermore, that the organisational culture lets older employees have the possibility of developing skills and be included in the development and new projects in the workplace regardless of age. Rotation of work tasks could be a way to learn new skills and abilities to keep employees employable in the organisation, but also in the case of reorganisations and change in the production. Rotation of tasks, e.g., changing occupation and activities within the organisation and switching work tasks was also suggested to make change of duties, to reduce monotony in tasks, and to increase motivation and job satisfaction. To utilise the senior employees’ experience-based knowledge by asking them to mentor new employees is a way of exchanging knowledge between generations and was described as a measure to increase the motivation and meaningfulness at work, but also to increase the employees’ employability and total know-how within the organisation.

## 4. Discussion

The population is ageing in many countries, therefore the aim of this study was to investigate organisational actions and proposals that promote an extended working life and maintain employability. The results of the conducted interviews in this study presented several organisational measures and suggestions to make improvements and to promote a healthy and sustainable working life for all ages in an extended working life, which also aligns with the nine determinant areas in the swAge-model [5,6,7]. The findings from this study contribute to strengthening the robustness of the theoretical implications and content of the swAge-model, theories on employability and how to promote a sustainable extended working life. 

The results collected from the interview data were organised into four main themes based on the four spheres of determination in the theoretical swAge-model, and with a number of extracted sub-themes of organisational measures and suggestions for a healthy and sustainable extended working life. The four main themes were: Measures for health effects associated with working environment; Measures for personal financial security; Measures for social inclusion and social support in the work situation; Measures for creativity and intrinsic work motivation. These themes are also closely related to the research on what people consider in the decision whether to keep working for some years or to retire, i.e., the consideration of: (i) their own health in relation to the work situation and work environment versus retirement; (ii) their personal financial situation in employment versus retirement; (iii) the opportunities of social inclusion in working life situations versus retirement; (iv) and the opportunities for meaningful and self-crediting activities in working life versus retirement [5,6,15]. 

The research results supported by theories of sustainable working life for all ages and employability [4,5,6,7,8,9,10,11,12,13,14,15,16] can hopefully help narrow the gap between theory and practice through the implications to be used as a toolbox to be applied by practitioners, managers and employees in their dialogue and discussion on employability and employee career development. Some measure activities and suggestions are sorted into more than one of the four main themes due to the context in the theme, e.g., rotation was described to be a good measure for different reasons. Therefore, some measure activities are described repeatedly in different themes based on separate reasons. However, the fact that a measure suggestion occurs based on different reasons in different themes shows the usefulness of this action to increase employability and for a sustainable extended working life.

### 4.1. Measures for Health Effects Associated with Working Environment

The health effects of the work environment affect employees’ ability and willingness to work and participate in working life and thus also their employability [5,6]. Systematic work environment management is a part of the EU regulations (89/39/EEG) and is compulsory in all the participating countries in the EU. Employers, according to this, must systematically examine the workplace on a regular basis to eliminate risks in the work environment and to prevent employees from suffering ill health and injuries caused by work. According to a recent study, a proper systematic work environment management in the workplace is statistically significant and associated with employees being able to work until an older age [25]. Many interviewees in this study stated that an awareness of employee ageing was needed in their systematic work environment control scheme, because the risks differ in different age groups. The biological ageing, as well as a higher risk of developing chronic diseases associated with an older age also increase the risks in the work environment, additionally, senior employees in general have declining hearing, sight, reaction ability and speed, which increase the risks of injury in this age group [5,6,26,27,28]. The interviewees in this study stated that, to reduce the risk of developing health problems because of the working environment, the organisational culture needs to promote the use of ergonomic aids; rotation and change of work tasks to reduce tear and strain; regular physical activity to maintain bodily functions and to stay mentally and physically healthy; the importance of diet for (occupational) health. Additionally, they suggest a better use of the occupational health care to support and promote health and to prevent physical and mental injury, illness and stress. However, a recent study in Sweden stated that the occupational health care accessible through the workplace nowadays is not statistically significant and associated with whether the employees are able to work until 65 years of age or beyond, but that there was a potential to change this, if occupational health care starts focussing on the age perspective in the workplaces [25]. In a healthy and sustainable working life for all ages awareness of this is important, as well as to implement measures to promote the work environments’ health effects.

### 4.2. Measures for Personal Financial Security

Employability is important for the individuals’ opportunity to maintain sound personal finances; however, personal finances in turn affect whether the individual can and wants to work and participate in working life [5,6]. It is much better for employees’ wellbeing if they work because they want to, and not because they cannot afford to leave the workplace [29]. In this study many interviewees stated that employees who were injured or had a disease and were not fully productive, also had problems with their employability if they could no longer execute their work tasks, if they could not be moved to another part of the organisation and enterprise or if they could not be re-employed within a reorganisation. Additionally, the interviewees stated that employees were not as employable if their skills and knowledge were dated or they did not keep up with the technology developments or did not have the latest knowledge, resulting in not being able to execute their duties and tasks as expected or to be relocated within their organisation. Earlier studies also state the need for employees to stay employable in the objective of being able to participate in an extended working life and provide for themselves [30,31]. Likewise, earlier studies state the necessity for organisations and enterprises to implement actions and measures to help the employees stay employable on the basis of ethics and humanity, and because risk prevention and competence development for the employees mean less expenses for the organisation compared to recruitment and new hiring [30,32]. In a healthy and sustainable working life for all ages it is important to have awareness and measures to promote personal financial security. 

### 4.3. Measures to Promote Social Inclusion, Participation, Coherence and Social Support in the Work Situation

The social support and social inclusion that the employees experience affect their ability and willingness to work and participate in working life, as well as their employability [5,6]. Many interviewees stated the importance of measures to promote social inclusion and social support in the work situation by considering whether all employees were included in the work social environment and in the work team. Additionally, earlier research states the importance of social inclusion, participation, coherence and social support in the work situation [15,33,34]. An important part in this area is the leadership. The manager needs to know that the employee needs to acknowledge his/her unique personal situation based on his/her: health, physical and mental working environment, need for recuperation, personal social/family situation, need for skills and knowledge, in relation to his/her work tasks, etc. Different employees need different social support and level of participation in the work situation. No employee can experience wellbeing or be able to produce at his/her best level if they feel excluded or discriminated against based on their age or other factors. Several interviewees describe the importance of actions and measures to decrease negative attitudes and (age) discrimination, and to promote and increase social support and senior employees’ self-esteem in the organisation. Many manager interviewees participating in this study stated that the older employees who were also mentors had a calming effect on the work team, which was an important productivity investment. Furthermore, earlier studies state the older employees’ experience and calming effect as important [28,33]. However, in order to increase the willingness to participate in an extended working life the senior employees also need to know they are valued, feel appreciated and have the possibility of participating and being included in a social context in the organisation. In a healthy and sustainable working life for all ages it is important to be aware of and take actions and measures to promote social inclusion, participation, coherence and social support in the work situation.

### 4.4. Measures to Promote Knowledge, Development, Creativity and Intrinsic Motivation in the Performance of Work Tasks

Whether employees are able to perform their work duties and activities or not affects their ability and willingness to work and participate in working life, as well as their employability [5,6]. According to many of the interviewees, motivation and skills were important in order to stay at work and in employment until an older age. Earlier studies also state the importance of skills, knowledge development, stimulation, meaningfulness and motivation in the work tasks for the employees’ willingness to work [35,36]. Additionally, earlier research states that the employees’ skills and knowledge are especially important to maintain their employability [30,31,36]. However, the organisational culture should afford older employees’ the possibility of developing skills and be included in the development and in new projects at the workplace, regardless of age, and to fuel the employees’ ability to keep up to date on their knowledge and experience of stimulation in work tasks. Measures and activities such as rotation of work tasks, e.g., changing occupation and activities within the organisation and switching work tasks, could be a way to reduce monotony in work tasks, to learn new skills and work tasks and to increase motivation and job satisfaction to retain employees in the organisation or within reorganisations and changes in production. Some interviewees suggested, and some had also tried, to utilise the older employees’ knowledge and experience through the possibility of being a mentor to new employees, by having responsibility for work areas, and by participating in activities such as discussions groups to exchange knowledge between generations of employees and increase the work teams’ common knowledge. In a healthy and sustainable working life for all ages it is important to be aware of and take measures to promote knowledge, development, creativity and the employees’ experience of intrinsic motivation in the performance of work tasks.

### 4.5. Limitations

A limitation of interview studies might be the established context, since researchers could approach the data with bias simply by their personality and body language in the interview situation [22]. However, the researchers who conducted these interviews are very used to interviewing and this limitation is therefore probably not an issue that affects the results. Additionally, there might be a risk that in direct content analysis the findings are more supportive rather than non-supportive of the pre-defined categories. However, in the present study, the pre-defined categories from the theoretical swAge-model were not used to guide data collection or probe questions during the interviews. The pre-defined categories were only applied during the analytical process. In terms of trustworthiness, the quality of a content analysis depends on the availability of extensive and appropriate data [37]. These findings represent different work domains, different occupational groups and professions, from different hierarchical positions within the organisations. However, findings from qualitative studies could not be stated to be transferable to all other cultural contexts. Regarding the transferability of the findings, it could be said that the themes are relevant to many others in similar workplaces. The findings were also well-aligned with those of earlier studies about employability and senior employees’ problems and possibilities regarding whether they can and want to participate in an extended working life, and thereby add to the accumulation of results regarding senior employees’ possibility of working in an extended working life [33].

### 4.6. Implications

The implications of the findings in this study, based on the interviewees’ own words, are that there are many possible measures to implement, which can be improved in organisations and enterprises to make circumstances in working life healthier and more sustainable for employees of all ages and to increase employability. One management tool to investigate what measures the individual employees primarily need to stay employable is through recurring individual career development discussions with employees. The manager and employee can systematically use a matrix together at the development discussion, where the needs are mapped and followed up at the subsequent development discussion (see Appendix A). The suggestions for organisational measures based on the findings in this study are:

#### 4.6.1. Measures to Promote a Sound Physical and Mental Work Environment for Employees by Considering to

Establish an organisational culture that promotes the use of technical and ergonomic aids.Establish reduced work stress, balanced workload, duties and responsibilities.Establish increased effort/reward balance.Establish good communication, information and participation to reduce work stress.Establish safety activities to reduce violence and threats in the work situation.Establish rotation, variation and change of duties and work tasks to reduce physical demand, static workload, wear and tear. As well as to reduce mental demands, stress, vulnerability and threatening situations.Establish work schedules that allow for sufficient recovery through breaks during work and time for recuperation between work shifts.Establish work pace at a reasonable rate so the employees can cope physically and mentally, if there are shifts with a faster work pace these should be complemented by shifts with a slower work pace.Establish physical activity for maintenance of the body, and to sustain and improve mental and physical good health.Establish the importance of a healthy diet for a good (occupational) health.Establish access to occupational health services and support to prevent work environment problems and increase good occupational health.

#### 4.6.2. Measures to Promote Personal Financial and Social Security by Considering to

Establish salary and financial benefits responding to a reasonable personal financial security and sufficient wellbeing.Establish work environmental security management and risk assessment to reduce the risk of health problems causing decreased employability and to reduce the risk of less financial benefits by sick leave, disability pension and unemployment.Establish measures for competence development and continued employability to make senior employees able to provide for themselves by working until an older age.

#### 4.6.3. Measures to Promote Social Inclusion and Social Support in the Work Situation by Considering to

Establish situational- and age-adapted leadership with a focus based on employees’ needs; these should be adapted for the best possible support of the work and development.Establish activities for social inclusion in the work team and to build a community and sense of security to increase the attraction of extended employment.Establish risk assessment and activities to reduce negative attitudes, victimisation, scapegoat mentality and (age) discrimination in the workplace.Establish social support to increase the senior employees’ self-esteem, sense of acknowledgement, sense of security and inclusion in the organisation.Establish access to a balanced level of information regarding ongoing activities and changes in the organisation.Establish participation in decisions that affect the employees’ work tasks and work situation.Establish activities to pay attention to individual employees as an important productivity investment, e.g., mentoring and calming effect on the group as well as entrepreneurship and ideas.Establish working schedules (with a balance between shorter and longer work shifts and time for recuperation between work shifts) that take into consideration and utilise the employees’ individual needs, participation and activities with family, leisure time and hobbies.

#### 4.6.4. Measures to Promote Creativity and Intrinsic Work Motivation by Considering to

Establish activities to increase motivation, stimulation and meaningfulness in work tasks.Establish the possibility for professional development, and new knowledge and competence development regardless of the employees’ age and position in the organisation.Establish an organisational culture that enables employees’ opportunities and skills (outside work as well) to be part of or to be involved in problem-solving, organisational development and new projects.Establish an organisational culture that takes into consideration and appreciates (senior) employees’ experience and knowledge as a production asset.Establish activities and an organisational culture that state the need for the employees’ continuous development of knowledge and skills to maintain their ongoing employability.Establish possibilities for the employees to take care of and manage larger or smaller work areas with the intention of stimulating, paying significance to and instilling motivation in the work tasks as well as increasing the individuals’ employability.Establish possibilities of rotation and change of duties and work tasks to reduce boredom and increase motivation and job satisfaction.

## 5. Conclusions

The demographic changes in society place strain on organisations and enterprises. The need for an extended working life, and the simultaneous higher average age among employees increase the need for measures to enable a healthy and sustainable working life for all ages. This study has identified various proposals for measures and actions that could increase the employees’ employability and ability to cope in working life until an older age. The identified issues are explained in association to each of the four themes’ context based on the spheres of determination in the theoretical swAge-model [5,6,7]. The results state that some needed measure activities and suggestions could be applied to more than one theme and issue, since working life is complex and the areas of determination associated with employability until an older age have some overlapping domains. However, to some extent, all nine determinant areas are associated with the four spheres of determination in the swAge-model and need to be considered in work places and societies that want working life to be satisfying and sustainable for employees of all ages in the work force. Employability is often perceived as a society’s and employer’s considerations of an employee’s ability to execute work tasks; however, it also regards whether employees want to and are able to work. Regular conversations, communication and close dialogue are needed and are prerequisites for good working conditions and a healthy working environment, as well as to be able to manage employees and develop the organisation further. The implication of the “tools for dialogue and discussion on employability and employees’ career development” (Appendix A) was based on the analysis of the interviews and findings in this study, and will hopefully be useful for organisations and enterprises in order to consider and implement needed measures with the intention of contributing to the employees’ employability in a sustainable working life for all ages. These measures should not only be applied to senior employees since all employees, hopefully, will become older employees some day and their health and skills need to be taken care of throughout their entire working life. 

A healthy and sustainable working life for all ages is highlighted as significant and needed around the world [1,2,3,4]. Additionally, the United Nations’ Sustainability Goal Agenda-2030 describes the need of decent work for all to enable a sustainable future [38]. The developed suggestions for measures and tools for dialogue and discussion in this study are intended to be implemented in workplaces. This study resulted in measure suggestions and a tool was developed based on the Swedish context, though hopefully the results can be usefully implemented and evaluated in depth in intervention studies of other countries’ work force context to develop a more sustainable working life for all ages. 

## Figures and Tables

**Table 1 ijerph-18-05626-t001:** Distribution of the study population.

Participant Group	Number of Participants			Number of Focus Group Interviews within the Group	Number of Individual Interviews within the Group
	Total	Women	Men		
Employees 55–72 years	87	42	45	14	22
First line managers	45	26	19	8	12
Trade union employees (two of whom were also safety delegates)	6	2	4	3	
Human resources personnel	7	3	4	3	
Total	145	73	72	28	34

## Data Availability

The data used in this study are managed by the authors. To access these data please contact the authors.

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
