# Peer review of "Organisational Measures and Strategies for a Healthy and Sustainable Extended Working Life and Employability—A Deductive Content Analysis with Data Including Employees, First Line Managers, Trade Union Representatives and HR-Practitioners"

_ijerph, 2021, doi:10.3390/ijerph18115626_

Round 1
Reviewer 1 Report
This paper is designed to promote the extended working life and maintain employability, and find out organisational actions and proposals to take improvements on healthy and sustainable working life for all ages in an extended working life. It is a well-designed and well-defined article and I recommend publishing after minor revision.
More detailed problems of this paper are listed as follows:
- Too many words for Keywords. We recommend to concise them and something more critical about the MS purposes.
- Given several mentions about the swAge-model, it is recommended that authors could provide more details about it for better understanding.
- Introduce the abbreviation before use like HR.
- Line 132: It is recommended that the title "Materials and Methods" is replaced by "Methods".
- We recommend to use "to" instead of "in order to".
- Line 13, 385, 483, 559: The word “make” doesn’t seem to fit. Consider replacing it with “take”.
- Line 142: Delete the extra space for “male- dominated”.
- Line 168 and 173: Repalce “in the study” by “the study”.
- Line 249: Consider removing the unnecessary “actually”.
- Line 270, 273, 277, 844, 899 and 967: Consider removing the unnecessary “own”.
- Line 296, 839 and 866: We recommend to use "workplace" instead of "work place".
- Line 475: Consider changing “of” with “for”.
- Line 1116: Delete the redundant “of”.
- Line 1154: Repalce “acknowledgement” by “acknowledgment”.
- Conclusion: Revise and make it more concise. This section should highlight the best results obtained in this study, as well as the prospects for future work.
Author Response
Too many words for Keywords. We recommend to concise them and something more critical about the MS purposes.
- The number of key words has been reduced
Given several mentions about the swAge-model, it is recommended that authors could provide more details about it for better understanding.
- The swAge-model is now better described in the introduction chapter.
Introduce the abbreviation before use like HR.
- This is now included
Line 132: It is recommended that the title "Materials and Methods" is replaced by "Methods".
We recommend to use "to" instead of "in order to".
Line 13, 385, 483, 559: The word “make” doesn’t seem to fit. Consider replacing it with “take”.
Line 142: Delete the extra space for “male- dominated”.
Line 168 and 173: Repalce “in the study” by “the study”.
Line 249: Consider removing the unnecessary “actually”.
Line 270, 273, 277, 844, 899 and 967: Consider removing the unnecessary “own”.
Line 296, 839 and 866: We recommend to use "workplace" instead of "work place".
Line 475: Consider changing “of” with “for”.
Line 1116: Delete the redundant “of”.
Line 1154: Repalce “acknowledgement” by “acknowledgment”.
- Thanks for the improvement suggestions! The language has now improved
Conclusion: Revise and make it more concise. This section should highlight the best results obtained in this study, as well as the prospects for future work.
- The conclusion chapter is now rewritten
Too many words for Keywords. We recommend to concise them and something more critical about the MS purposes.
- The number of key words has been reduced
Given several mentions about the swAge-model, it is recommended that authors could provide more details about it for better understanding.
- The swAge-model is now better described in the introduction chapter.
Introduce the abbreviation before use like HR.
- This is now included
Line 132: It is recommended that the title "Materials and Methods" is replaced by "Methods".
We recommend to use "to" instead of "in order to".
Line 13, 385, 483, 559: The word “make” doesn’t seem to fit. Consider replacing it with “take”.
Line 142: Delete the extra space for “male- dominated”.
Line 168 and 173: Repalce “in the study” by “the study”.
Line 249: Consider removing the unnecessary “actually”.
Line 270, 273, 277, 844, 899 and 967: Consider removing the unnecessary “own”.
Line 296, 839 and 866: We recommend to use "workplace" instead of "work place".
Line 475: Consider changing “of” with “for”.
Line 1116: Delete the redundant “of”.
Line 1154: Repalce “acknowledgement” by “acknowledgment”.
- Thanks for the improvement suggestions! The language has now improved
Conclusion: Revise and make it more concise. This section should highlight the best results obtained in this study, as well as the prospects for future work.
- The conclusion chapter is now rewritten
Reviewer 2 Report
1) I have reviewed this manuscript and made some editing suggestions, mostly in terms of the English language and syntax. These suggestions are included in the original manuscript (PDF document which I attach as the support document) as deletions, highlighted text, insertions, requests for clarification, and sticky notes where required.
2) The paper is fairly long and there is some repetition of information that would benefit from a revision to streamline what is presented, particularly in the 'Introduction' and 'Results' section. The repetition and length of the paper make it somewhat tedious to read and process the information that is presented.
3) Too many keywords have been given, some of which do not appear in the main text. The author should consider shortening the list of keywords to include only the most appropriate ones, and to avoid repetition of similar terms.
4) 'Author Contributions: The author has completed the analysis, authored the paper and approved the final manuscript'. Prior to finalising and submitting the manuscript for publication consideration, the sole author would benefit from requesting input from a professional in the same field to review the study design and the findings, so as to have a second opinion about what is stated by the author.
5) The author has not indicated when the interviews were carried out and the data collected (time frame). Is this a completely new study or have the data obtained and processed for this study population been presented partially in another publication?

Author Response
1) I have reviewed this manuscript and made some editing suggestions, mostly in terms of the English language and syntax. These suggestions are included in the original manuscript (PDF document which I attach as the support document) as deletions, highlighted text, insertions, requests for clarification, and sticky notes where required.
- Thank you so very much for your very good suggestions for improvements to the text! We have accepted and followed these proposals.
2) The paper is fairly long and there is some repetition of information that would benefit from a revision to streamline what is presented, particularly in the 'Introduction' and 'Results' section. The repetition and length of the paper make it somewhat tedious to read and process the information that is presented.
- What affects employability and that individuals can and want to work is very complex and spans a large and interdisciplinary field of research. The paper is handling overlapping domains and deal with the identification of different categories. We therefore see it as risking neglecting this complexity and the need for the overall picture to be able to take comprehensive action if we are forced to remove and shorten the article.
3) Too many keywords have been given, some of which do not appear in the main text. The author should consider shortening the list of keywords to include only the most appropriate ones, and to avoid repetition of similar terms.
- The list of key words is now shorter.
4) 'Author Contributions: The author has completed the analysis, authored the paper and approved the final manuscript'. Prior to finalising and submitting the manuscript for publication consideration, the sole author would benefit from requesting input from a professional in the same field to review the study design and the findings, so as to have a second opinion about what is stated by the author.
- The researcher who participated in the analysis work, but who was not involved in the actual writing of this script, has now also been included as a co-author to show that the main author has not done the analysis work alone.
5) The author has not indicated when the interviews were carried out and the data collected (time frame). Is this a completely new study or have the data obtained and processed for this study population been presented partially in another publication?
- The participants were recruited in different ways in year 2011-2020. Some of the collected data has been has previously been analysed in direction of other purposes, i.e. why some had left working life early and others worked to old age, work motivation, the attitude between managers and employees, and the transfer of knowledge between generations
Reviewer 3 Report
The article addresses a very interesting topic, applies an appropriate methodology and presents the results clearly. The purpose of the study is described in a logical, comprehensible, and explicit manner. A notable argumentation in support of research pointing to gaps in the literature is established. The significance of the research is clearly established. Also, the authors presented the limitations of the research and highlighted the implications of this research.
However, the authors should also consider the following suggestions:
Abstract
- I think that the abstract does not fall within the recommendation of a single paragraph of about 200 words maximum. Also, I would suggest to the author to use the style of structured abstract, but without mentioning the headings (e.g. background, method, results, conclusions).
Keywords
- I consider that the list of keywords is too long. I would suggest to the author to list three to ten pertinent keywords specific to the article yet reasonably common within the subject discipline.
2. Materials and Methods
2.2. Data collection
- “Data collection” section should be improved with information about the research period. When were focus group and individual interviews applied?
2.3. Analysis
-Line 200: The author mentioned that “All the interviews were analysed together through text analysis method”. Was a specific text analysis tool or software used?
5. Conclusion
- I would suggest to the author to complete this section with information about the future directions of research, if they have been identified.
6. Patents/appendix 1
- In the appendices, Figures, Tables, etc. should be labeled starting with “A”—e.g., Figure A1, Figure A2, etc.
References
- The “References” section should be reviewed in accordance with the manuscript preparation template of the International Journal of Environmental Research and Public Health (IJERPH). Example:
- For articles published in journals, the year of publication should be highlighted in bold and the journal name should be written in italics.
- For book, the Book title should be written in italics.
- For source no. 2 the DOI is mentioned “https://doi.org/10.1787/b6d3dcfc-en”, and for source no.5 otherwise: “doi: 1016/j.apergo.2020.103082”. I would suggest to the author to write the DOI in the same way in the entire references. Thus, the author can decide and write DOI either: https://doi.org/10.1787/b6d3dcfc-en or DOI: 10.1787/b6d3dcfc-en.
- Web sources should be mentioned as follows: Title of Site. Available online: URL (accessed on Day Month Year).
Author Response
- I think that the abstract does not fall within the recommendation of a single paragraph of about 200 words maximum. Also, I would suggest to the author to use the style of structured abstract, but without mentioning the headings (e.g. background, method, results, conclusions).
- The abstract is now shorter, and the headings is removed.
Keywords
- I consider that the list of keywords is too long. I would suggest to the author to list three to ten pertinent keywords specific to the article yet reasonably common within the subject discipline.
- The list of key words is now shorter.
Materials and Methods
Data collection
- “Data collection” section should be improved with information about the research period. When were focus group and individual interviews applied?
- The data collection and the interviews were done over the year 2011-2020. This is now included.
Analysis
-Line 200: The author mentioned that “All the interviews were analysed together through text analysis method”. Was a specific text analysis tool or software used?
- The text analysis was performed traditionally and not with any software program.
Conclusion
- I would suggest to the author to complete this section with information about the future directions of research, if they have been identified.
- This is now better described.
Patents/appendix 1
- In the appendices, Figures, Tables, etc. should be labeled starting with “A”—e.g., Figure A1, Figure A2, etc.
- This is now done
References
- The “References” section should be reviewed in accordance with the manuscript preparation template of the International Journal of Environmental Research and Public Health (IJERPH). Example:
- For articles published in journals, the year of publication should be highlighted in bold and the journal name should be written in italics.
- For book, the Book title should be written in italics.
- For source no. 2 the DOI is mentioned “https://doi.org/10.1787/b6d3dcfc-en”, and for source no.5 otherwise: “doi: 1016/j.apergo.2020.103082”. I would suggest to the author to write the DOI in the same way in the entire references. Thus, the author can decide and write DOI either: https://doi.org/10.1787/b6d3dcfc-en or DOI: 10.1787/b6d3dcfc-en.
- Web sources should be mentioned as follows: Title of Site. Available online: URL (accessed on Day Month Year).
- The reference is now revised in accordance with the manuscript preparation template
Reviewer 4 Report
Some clarification is needed about whether the 4 spheres of determinants of the SwAge model can have overlapping domains as with the four themes of this article, i.e.
Measures for the work environment's health effects; Measures for personal financial security; Measures for social support and inclusion; Measures for execution of work tasks.
For example, one could imagine if someone has more social support, then personal financial security could be less stressful. Similarly, the work environments if needs to be in an area of relative isolation could mean less ability to have social support from family members who are far away from the location of work. How would the color-coding analyses handle such overlapping domains and deal with the identification of different categories?
Since there are several places where summary is included in the text, e.g. a summary was found in sections: 3.1.11. 3.2.5 3.3.7 3.4.7
It may be a good idea to have a bit more details in the heading to distinguish these various summary sections, e.g.
3.1.11 summary about work environments health effects
3.2.5 summary about personal financial security
3.3.7 Summary about social support and inclusion and
3.4.7 summary about the execution of work tasks
Author Response
Some clarification is needed about whether the 4 spheres of determinants of the SwAge model can have overlapping domains as with the four themes of this article, i.e. Measures for the work environment's health effects; Measures for personal financial security; Measures for social support and inclusion; Measures for execution of work tasks. For example, one could imagine if someone has more social support, then personal financial security could be less stressful. Similarly, the work environments if needs to be in an area of relative isolation could mean less ability to have social support from family members who are far away from the location of work. How would the color-coding analyses handle such overlapping domains and deal with the identification of different categories?
- Thank you for this reflection. This was earlier already described in the limitation chapter, but is now also described in the method chapter.
Since there are several places where summary is included in the text, e.g. a summary was found in sections: 3.1.11. 3.2.5 3.3.7 3.4.7. It may be a good idea to have a bit more details in the heading to distinguish these various summary sections, e.g.
3.1.11 summary about work environments health effects
3.2.5 summary about personal financial security
3.3.7 Summary about social support and inclusion and
3.4.7 summary about the execution of work tasks
- Thank you for this improvement suggestion! This is now done.
Reviewer 5 Report
The topic of the paper is very interesting. But the study justification, research methodology, and process seem to be very weak. I felt like this paper was written as a technical report rather than an academic paper. Also, as qualitative research needs rigorous processes and techniques, the research framework is very important. But this paper does not provide these processes and techniques in a detailed manner. Although the research findings should be related to the literature review, in addition, it is not in the paper (i.e., this manuscript does not include the literature review section). More seriously, to formulate some implications, any statistical techniques and/or frameworks should have been used rather than simply describing what the participants said even though it was a qualitative study. I would suggest the author(s) to submit their manuscript to an industry-focused journal rather than an academic journal for the above reasons.
Author Response
Thank you for your suggestions!
However, this qualitative study followed the usually design and a well-known analyse method, and that does not include any statistical techniques.
Round 2
Reviewer 2 Report
I have reviewed the second version of the manuscript. I continue to be of the opinion that the paper is very long and quite repetitive across various sections. However, the authors have endeavoured to address some of the concerns that I raised in my review of the first version of the manuscript.
I have still had to make editing corrections in this second version, mostly for language and syntax - please see my sticky notes and comments in the PDF of the second version, which I attach here for your review.

Author Response
Thank you so much for your valuable comments!
We have now considered these and made changes based on the proposals.
Sincerely, professor Kerstin Nilsson
Reviewer 5 Report
I appreciate the amount of work that has gone through these revisions. However, I am still worried about this study's methods and lack of the literature review section. I know this review is disappointing. I wish the authors all the best as they continue their research in this area and in all aspects of their works.
Author Response
Thank you very much for your valuable comments!
Sincerely, professor Kerstin Nilsson